# Associations of active travel with adiposity among children and socioeconomic differentials: a longitudinal study

Anthony A Laverty ,[1] Thomas Hone ,[1] Anna Goodman,[2] Yvonne Kelly ,[3] Christopher Millett[1]

[1]Public Health Policy Evaluation Unit, School of Public Health, Imperial College London, London, UK
[2]Faculty of Epidemiology and Population Health, London School of Hygiene & Tropical Medicine, London, UK
[3]Epidemiology and Public Health, University College London, London, UK

**Correspondence to**
Dr Anthony A Laverty;
a.laverty@imperial.ac.uk

## ABSTRACT

**Objectives** Examine longitudinal associations between modes of travel to school and adiposity.

**Setting** The UK.

**Participants** 8432 children surveyed at ages 7, 11 and 14 years from the UK Millennium Cohort Study.

**Primary and secondary outcomes** Objective percentage body fat and body mass index (BMI). Transport mode was categorised as private motorised transport, public transport and active transport (walking or cycling). Socioeconomic position (SEP) was measured by household income group and occupational social class. We adjusted analyses for changes in the country of UK, frequency of eating breakfast, self-reported growth spurts, hours of screen time and days per week of moderate-to-vigorous physical activity. Longitudinal (panel) regression models adjusting for individual fixed effects examined associations in changes in mode of travel to school and adiposity, controlling for both time-varying and time-invariant potential confounders. Interaction tests and stratified analyses investigated differences by markers of SEP.

**Results** At age 14 years, 26.1% of children (2198) reported using private motorised transport, 35.3% (2979) used public transport and 38.6% (3255) used active transport to get to school. 36.6% (3083) of children changed mode two times between the three waves and 50.7% (4279) changed once. Compared with continuing to use private transport, switching to active transport was associated with a lower BMI (−0.21 kg/m$^2$, 95% CI −0.31 to −0.10) and body fat (−0.55%, 95% CI −0.80% to −0.31%). Switching to public transport was associated with lower percentage body fat (−0.43%, 95% CI −0.75% to −0.12%), but associations with BMI did not reach statistical significance (−0.13 kg/m$^2$, 95% CI −0.26 to 0.01). Interaction tests showed a trend for these effects to be stronger in more deprived groups, but these interactions did not reach statistical significance.

**Conclusion** This longitudinal study during a key life course period found switching to physically active forms of travel can have beneficial adiposity impacts; these associations may be more apparent for more disadvantaged children. Increasing active travel has potential to ameliorate inequalities.

## Strengths and limitations of this study

► We used an established and representative UK birth cohort study with high-quality data following up the same children over 8 years.
► We used two objective measures of adiposity and employed models that produce robust estimates of within-individual changes and were able to adjust for unobserved time-invariant individual characteristics.
► However, we rely on self-reported mode of travel to school, which did not capture information on multi-mode journeys or variation in different modes across the week.
► There was an average of 3 years between data collection points, and we do not know at what point between data collections individuals changed their travel behaviour.
► While we were able to consider a wide range of co-variates, there may be some residual confounding from time-varying factors not considered such as choosing to lead a healthier lifestyle in other ways also, and changes in adiposity may be occurring via alternative mechanisms than active travel.

## INTRODUCTION

Insufficient physical activity is associated with increased cardiometabolic risk, including higher levels of adiposity in children as well as adults.[1] Children are an important group for physical activity promotion and the majority of children and adolescents do not reach recommended levels.[2 3] Additionally evidence from both the UK and international studies suggests that levels of physical activity track across the life course, meaning that those who are insufficiently active early in life are at risk of this continuing into later life.[4–6] The period of transition from primary to secondary education is characterised by significant changes to children's social and physical environment which can strongly influence health behaviours.[7]

Active travel (that is, the use of non-motorised modes of travel) has the potential to increase physical activity and reduce adiposity.[8] The use of active travel in children has been linked to higher levels of physical activity in a range of settings, including during the journey to school.[6 9–11] Modes of travel change significantly during childhood, with the majority of travel being with caregivers in primary school and then changing to more independent modes later.[12] There are also significant differentials across socioeconomic position (SEP) in physical activity profiles, use of active travel and cardiometabolic health risks in children, which together contribute to levels of health inequalities between groups.[13–15] Inequalities are also present among children for other factors and behaviours which are associated with greater risks of obesity and ill-health, including screen time and dietary behaviours.[16] While active travel may have the potential to offset some of these inequalities, a recent review by Sport England concluded that there is a comparative lack of evidence on the impacts of active travel across groups across markers of SEP.[17]

Much of the evidence on the impacts of active travel has been cross-sectional and there has been a lack of evidence using large-scale prospective studies, especially in children. Studies have highlighted conflicting evidence on the potential role of active travel to school on body composition and adiposity, and a recent systematic review found a limited number of prospective studies in children and did not identify an association between active travel and obesity in children.[18–21] This study adds to the evidence base by using a robust analytical strategy on data from a representative cohort of children during a key period in their development to investigate the role of active travel to school on objectively measured adiposity, and the potential for differential effects according to two different measures of SEP. Multivariate longitudinal (panel) regression models are employed to follow the same individuals over time, adjust for time invariant factors, and assess changes in transport usage and health outcomes.

## METHODS
### Sample and data
Data for this study came from the Millennium Cohort Study (MCS), which is a birth cohort study in the UK of children born between September 2000 and January 2002, sampled at approximately 9 months old.[22] Children were initially sampled in the year 2000 and follow-up data are currently available from six sweeps. Sampling was based on a cluster-stratified framework with smaller population groups being oversampled, including those in the smaller nations of the UK, as well as those in disadvantaged areas and from ethnic minority backgrounds.

These analyses use data from the fourth sweep of data collection when children were aged on average 7 years old (response rate 82% and age range 6–8 years); on the fifth sweep of data collection, children were on average 11 years old (response rate 81% and age range 10–12 years); and on the sixth sweep of data when aged on average 14 years old (response rate 76%, age range 13–15 years). A range of topics covering the economic, social and health circumstances of children are covered alongside objective adiposity measurements. We focused on these sweeps of data as levels of active travel were low before the age of 7 years. Questionnaires and measures are administered by trained interviewers to standardised protocols and further details are available from http://www.cls.ioe.ac.uk/.[22]

### Variables
This study used two objective measures of adiposity at all three time points: body mass index (BMI) and percentage body fat. Height was measured using a Leicester stadiometer, and weight and body fat were measured using Tanita BF-522 W scales.[22] This provides objective data on BMI, rather than relying on participants self-reporting their height and weight.

The exposure for this study was mode of travel to school assessed by asking caregivers 'How does (child) usually travel to school?', with prompts to give the mode with longest distance if more than one mode was used. There were seven potential responses: public transport; school or local authority bus; minibus or coach; car or other vehicle (including taxi); bicycle; walking and other. We categorised these responses into three categories of private motorised (car or other vehicle (including taxi)), public transport (public transport; school or local authority bus; minibus or coach) and active travel (walking and cycling). We combined walking and cycling into one overall category due to low numbers of cycling: 76 at age 7 years, 187 at age 11 years and 147 at age 14 years. We excluded 'other' responses (253 at age 14 years (2.2%), 1203 at age 11 years (9.0%), 174 at age 7 years (1.3%)) due to uncertainty about what modes of travel these may be and the fact that they likely represent a heterogeneous mix of modes.

A range of covariates were included as potential confounders of the relationship between travel mode and adiposity, covering socioeconomic factors and health behaviours. Included covariates from all ages were sex; country of residence (England, Wales, Northern Ireland, Scotland); self-reported ethnicity; portions of fruit consumed per day (<3 vs ≥3 pieces per day); breakfast consumption (7 days per week vs not); screen use (hours per weekday watching TV or using a computer categorised as <3 hours vs ≥3 hours). We also created a variable to reflect days per week of sport/exercise each week (categorised as <once a week, one to two times a week, more than three times a week). We also used data on whether caregivers report that their child has experienced a growth spurt, assessed using the question 'Most children have a growth spurt as they approach and during their teens. Would you say that (cohort child's name)'s growth spurt…?' This question was asked only at ages 11 and 14 years and we categorised this into two categories of has definitely started versus not.

We used two markers of SEP both ascertained by self-report: household income group (equivalised based on the Organisation for Economic Co-operation and Development and categorised in five groups) and household occupational social class measured using the National Statistics Socio-Economic Classification (NS-SEC; categorised as not economically active; lower; intermediate; managerial/professional).

For longitudinal analyses, we included children who had complete data available from ages 7 to 14 years. For cross-sectional analyses at age 14 years, there were 11 714 children in the sample, of which we used data from 10 722 (91.5%) after excluding 253 (2.2%) for using an 'other' form of transport and 739 (6.3%) for missing data on relevant covariates. For longitudinal analyses, we include data from 8432 children (82.5% of those with complete data at age 14 years) who had data on all relevant data from ages 7 to 14 years.

## Analyses

The distribution of modes of travel to school across sociodemographic and behavioural characteristics was first examined descriptively.

Multivariate longitudinal (panel) regression models were used to assess multiple waves of data for each child. These models allow individuals to be followed over time, and also account for the clustered nature of the data (ie, multiple responses per child over time).[23] Linear specifications were chosen due to the continuous nature of the variables of interest. Fixed-effects specifications were chosen to adjust for individual time-invariant characteristics of individuals. These for example, could be sex, genetic factors, unmeasured sociodemographic factors, early life determinants, or personality traits that could bias this relationship between transport and health. As these individual fixed effects were adjusted for, the models exploit the within-individual variation over time and measure the association between changes in travel mode and changes in adiposity while controlling for all (measured and unmeasured) time-invariant characteristics.[23 24] We additionally controlled for time-variant confounders in the models that could bias this association over time. Confounders we controlled for were: changes in country; highest NS-SEC in household; household income; eating breakfast; frequency of eating breakfast; self-reported growth spurt; hours of TV and computer use per day and days per week of moderate-to-vigorous physical activity. Additionally time fixed effects were adjusted for dummies for each wave of data to account for changes over time. The reported effect sizes from these models are interpreted as the average association between changing travel mode across all three time points compared with changes among participants who used private transport.

First, this model was used for the two separate outcomes of BMI and percentage body fat. Second to assess differences across our two markers of SEP, we used interaction tests to assess if differences were statistically different and performed stratified analyses. All analyses used cluster robust SEs to account for heteroskedasticity and autocorrelation, and employed survey weights to correct for differential response rates between groups and over time.[22 23] We performed the Hausman test of random versus fixed-effect models, which indicated (p<0.001) that fixed effects were superior. Analyses were conducted using Stata V.14.0 and implemented using the xtreg package for longitudinal analysis.

## Exploratory analyses

In exploratory analyses we also investigated the potential that any differentials across SEP could be caused by differing duration of journeys to school, which may both determine mode of transport and be a proxy for amounts of physical activity involved in journeys. We restricted these analyses to individuals using physically active travel modes (walking or cycling to school). We used the question 'On a typical day, how long does it take (cohort member's name) to get from home to school, one way?' with options '<5 min, 5–10 min, 15–30 min, 30–45 min, 45 min–1 hour, 1–2 hours, 2 hours or more'. We have categorised this into a binary variable of 15 min or more versus less than 15 min each way, as a proxy for journey distance. We conducted logistic regression to assess which participants were more likely to travel 15 min or more to school at age 14 years.

## Patient and public involvement

Patients were not involved in the design of this study.

## RESULTS

Details of the sample and their characteristics at age 14 years are given in table 1. Of these, 26.1% of children used private motorised modes to travel to school, 35.3% used public transport and 38.6% used active modes. Levels of active travel ranged from 45.1% in the lower NS-SEC groups to 35.2% among the managerial/professional NS-SEC group, and were highest among those in the lowest income group (46.0% vs 36.2% among those in the highest income households). The majority of children (5409, 64.1%) lived in England, were white (7329, 86.9%) and had more than 3 hours per day of screen use (6535, 77.5%). Modes of travel to school at previous ages are shown in online supplemental tables 1 and 2. These show, for example, that 47.3% of children used private motorised modes to travel to school at age 7 years, compared with 43.7% at age 11 years. Levels of change in transport mode were high: 36.6% (3083) of children changed mode two times between the three waves; 50.7% (4279) changed once and 12.7% (1070) did not change travel mode. Overall switching between modes was more common between ages 11 and 14 years (4340 participants (51.5% of sample) changed travel mode), than between ages 7 and 11 years (2079 participants (24.7% of sample)).

Results from fixed-effects longitudinal regression models are given in table 2. BMI increased substantially between the ages of 7 and 14 years (+4.59 kg/m², 95% CI

**Table 1** Mode of travel to school in the Millennium Cohort Study in 2014/2015, age 14/15 years

| | | Number | Car % (number) | PT % (number) | Active % (number) |
|---|---|---|---|---|---|
| Gender | Male | 4102 | 24.2 (992) | 35.4 (1453) | 40.4 (1657) |
| | Female | 4330 | 27.9 (1206) | 35.2 (1526) | 36.9 (1598) |
| Country | England | 5409 | 26.9 (1455) | 29.8 (1612) | 43.3 (2342) |
| | Wales | 1182 | 25.5 (302) | 40.3 (476) | 34.2 (404) |
| | Scotland | 958 | 20.0 (192) | 40.5 (388) | 39.5 (378) |
| | Northern Ireland | 883 | 28.2 (249) | 57.0 (503) | 14.8 (131) |
| NS-SEC social class | Not economically active | 2015 | 26.4 (532) | 32.6 (657) | 41.0 (826) |
| | Lower | 1710 | 22.9 (392) | 32.0 (547) | 45.1 (771) |
| | Intermediate | 2054 | 26.5 (544) | 38.3 (787) | 35.2 (723) |
| | Managerial/professional | 2653 | 27.5 (730) | 37.2 (987) | 35.2 (934) |
| Household income group | Lowest | 1092 | 26.1 (285) | 27.9 (305) | 46 (502) |
| | Second lowest | 1286 | 23.2 (298) | 36.5 (469) | 40.4 (519) |
| | Middle | 1737 | 25.2 (438) | 37.2 (646) | 37.6 (653) |
| | Second highest | 2121 | 27.3 (578) | 35.7 (757) | 37.1 (786) |
| | Highest | 2196 | 27.3 (599) | 36.5 (802) | 36.2 (795) |
| Ethnic group | White | 7329 | 25 (1832) | 36.5 (2677) | 38.5 (2820) |
| | Mixed | 64 | 17.2 (11) | 45.3 (29) | 37.5 (24) |
| | Indian | 208 | 34.6 (72) | 26.4 (55) | 38.9 (81) |
| | Pakistani or Bangladeshi | 468 | 42.7 (200) | 10.9 (51) | 46.4 (217) |
| | Black or Black British | 241 | 18.7 (45) | 49.4 (119) | 32.0 (77) |
| | Other | 122 | 31.1 (38) | 39.3 (48) | 29.5 (36) |
| Growth spurt | Not yet/barely started | 2952 | 25.4 (750) | 35.9 (1060) | 38.7 (1142) |
| | Begun or completed | 5480 | 26.4 (1448) | 35.0 (1919) | 38.6 (2113) |
| Breakfast consumption | <7 days per week | 3824 | 25.8 (985) | 35.0 (1339) | 39.2 (1500) |
| | Every day | 4608 | 26.3 (1213) | 35.6 (1640) | 38.1 (1755) |
| Portions of fruit per day | ≤3 | 5775 | 26.1 (1510) | 34.5 (1994) | 39.3 (2271) |
| | ≥2 | 2655 | 25.9 (688) | 37.1 (985) | 37.0 (982) |
| Physical activity | ≤2 days per week | 2342 | 27.8 (650) | 34.3 (804) | 37.9 (888) |
| | 3–4 days per week | 2905 | 25.0 (725) | 37.6 (1092) | 37.5 (1088) |
| | ≥5 days per week | 3185 | 25.8 (823) | 34.0 (1083) | 40.2 (1279) |
| TV or computer use | ≤2 hours per day | 1897 | 26.4 (501) | 38.2 (724) | 35.4 (672) |
| | ≥3 hours per day | 6535 | 26.0 (1697) | 34.5 (2255) | 39.5 (2583) |
| Overall | | 8432 | 26.1 (2198) | 35.3 (2979) | 38.6 (3255) |

NS-SEC, National Statistics Socio-Economic Classification; PT, public transport.

4.45 to 4.73 for age 14 years compared with age 7 years). Caregiver-reported beginning of growth spurts between ages 11 and 14 years was associated with increased BMI (0.22 kg/m$^2$, 95% CI 0.14 to 0.31). Switching to eating breakfast every day was associated with lower BMI (−0.39 kg/m$^2$, 95% CI −0.51 to −0.27), compared with not eating breakfast every day, as was moving to doing physical activity 5 days or more per week (−0.32 kg/m$^2$, 95%

**Table 2** Results from longitudinal fixed-effects regression of impacts of switching school travel mode and adiposity

| | | BMI | | % body fat | |
|---|---|---|---|---|---|
| | | Coeff | 95% CI | Coeff | 95% CI |
| Mode of travel | Private transport | Ref | Ref | rRef | Ref |
| | Public transport | −0.13 | −0.26 to 0.01 | −0.43 | −0.75 to −0.12 |
| | Walk/cycle | −0.21 | −0.31 to −0.10 | −0.55 | −0.80 to −0.31 |
| Age (years) | 7 | Ref | Ref | Ref | Ref |
| | 11 | 2.44 | 2.37 to 2.52 | 1.06 | 0.87 to 1.25 |
| | 14 | 4.59 | 4.45 to 4.73 | 0.21 | −0.14 to 0.56 |
| Country | England | Ref | Ref | Ref | Ref |
| | Wales | 0.13 | −1.10 to 1.37 | 1.77 | −0.54 to 4.08 |
| | Scotland | 0.01 | −1.00 to 1.01 | 0.46 | −1.88 to 2.79 |
| | Northern Ireland | −0.10 | −1.92 to 1.72 | 1.07 | −3.04 to 5.19 |
| NS-SEC social class | Not economically active | Ref | Ref | Ref | Ref |
| | Lower | −0.04 | −0.18 to 0.10 | −0.22 | −0.53 to 0.10 |
| | Intermediate | −0.08 | −0.21 to 0.06 | −0.24 | −0.58 to 0.07 |
| | Managerial/professional | 0.08 | −0.06 to 0.22 | 0.16 | −0.18 to 0.50 |
| Household income group | Lowest | Ref | Ref | Ref | Ref |
| | Second lowest | 0.08 | −0.07 to 0.23 | 0.41 | 0.05 to 0.77 |
| | Middle | 0.13 | −0.03 to 0.29 | 0.36 | −0.02 to 0.75 |
| | Second highest | 0.06 | −0.11 to 0.23 | 0.12 | −0.30 to 0.54 |
| | Highest | −0.01 | −0.20 to 0.17 | 0.14 | −0.30 to 0.59 |
| Growth spurt | Not yet/barely started | Ref | Ref | Ref | Ref |
| | Begun or completed | 0.22 | 0.14 to 0.31 | 0.10 | −0.11 to 0.31 |
| Breakfast consumption | <7 days per week | Ref | Ref | Ref | Ref |
| | Every day | −0.39 | −0.51 to −0.27 | −1.34 | −1.62 to −1.06 |
| Portions of fruit per day | ≤3 | Ref | Ref | Ref | Ref |
| | ≥2 | 0.05 | −0.04 to 0.15 | 0.11 | −0.11 to 0.33 |
| Physical activity | ≤2 days per week | Ref | Ref | Ref | Ref |
| | 3–4 days per week | −0.06 | −0.16 to 0.04 | −0.25 | −0.48 to −0.01 |
| | ≥5 days per week | −0.32 | −0.41 to −0.23 | −1.04 | −1.25 to −0.82 |
| TV or computer use | ≤2 hours per day | Ref | Ref | Ref | Ref |
| | ≥3 hours per day | 0.07 | −0.02 to 0.16 | 0.22 | −0.01 to 0.44 |

Results from one separate model for BMI and one separate model for percentage body fat. Models adjusted for individual-level fixed effects and changes in: country; highest NS-SEC in household; household income; eating breakfast; frequency of eating breakfast; self-reported growth spurt; hours of TV and computer use per day and days per week of moderate-to-vigorous physical activity.
BMI, body mass index; Coeff, coefficient; NS-SEC, National Statistics Socio-Economic Classification; Ref, reference category.

CI −0.41 to −0.23). On average, after adjusting for these trends, switching from private to active travel was associated with a −0.21 kg/m$^2$ lower BMI (95% CI −0.31 to −0.10), while switching from private travel to public transport was associated with a −0.13 (95% CI −0.26 to 0.01) lower BMI.

Percentage body fat was higher at age 11 years (+1.06%, 95% CI 0.87% to 1.25%) but not at age 14 years (+0.21%, 95% CI −0.14% to 0.56%) than at age 7 years. Switching to eating breakfast 7 days a week was associated with a −1.34% lower body fat (95% CI −1.62% to −1.06%) but the effect of switching to 3 or more hours a day of screen use did not quite reach statistical significance (+0.22%, 95% CI −0.01% to 0.44%). Switching to 3–4 days a week of physical activity was associated with a −0.25% lower body fat (95% CI −0.48% to −0.01%) and switching to 5 or more days a week of physical activity a −1.04% lower body fat (95% CI −1.25% to −0.82%). Switching from a private mode to an active mode of travel was associated with a −0.55% lower percentage body fat (95% CI −0.80% to −0.31%) and switching to public transport with a −0.43% lower percentage body fat (95% CI −0.75% to −0.12%).

Associations of changes in travel mode with BMI across our markers of SEP are shown in figure 1 and

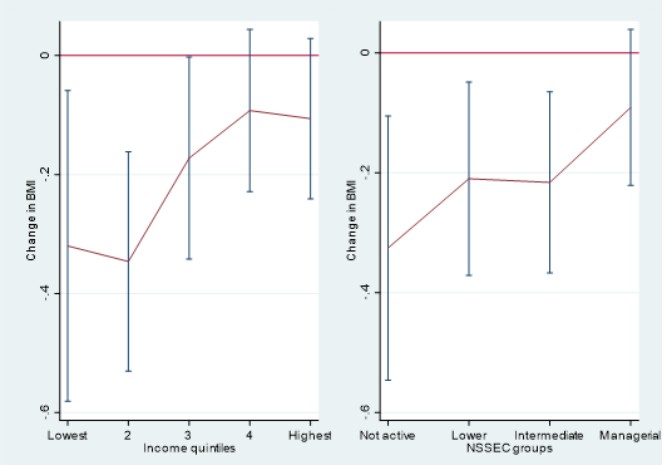

**Figure 1** Changes in BMI and 95% CIs by mode of travel to school. Results from longitudinal fixed-effects regression of impacts of switching school travel mode and BMI stratified by markers of socioeconomic position. These represent the average effect of switching to using active transport from any other mode. BMI, body mass index; NS-SEC, National Statistics Socio-Economic Classification.

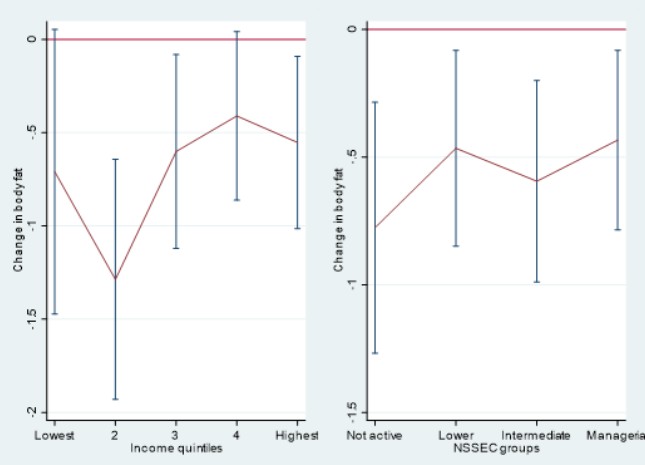

**Figure 2** Changes in % body fat and 95% CIs by mode of travel to school. Results from longitudinal fixed-effects regression of impacts of switching school travel mode and percentage body fat stratified by markers of socioeconomic position. These represent the average effect of switching to using active transport from any other mode. NS-SEC, National Statistics Socio-Economic Classification.

online supplemental table 3. Interaction tests did not reveal differences between income groups (p=0.130) and NS-SEC (p=0.074). However, visual interpretation of these charts did appear to show that these relationships were most concentrated in the lower SEP groups. For example, switching to active travel was associated with a $-0.32 \text{ kg/m}^2$ lower BMI (95% CI $-0.58$ to $-0.06$) among those in the lowest household income group compared with $-0.11 \text{ kg/m}^2$ among the highest household income group (95% CI $-0.24$ to 0.03). Similarly switching to active travel was associated with a $-0.33 \text{ kg/m}^2$ lower BMI (95% CI $-0.55$ to $-0.11$) among those in the not economically active NS-SEC group, but a $-0.09 \text{ kg/m}^2$ lower BMI (95% CI $-0.22$ to 0.04) among those in managerial/professional NS-SEC group.

Results were similar for percentage body fat and are shown in figure 2 and online supplemental table 4. Interaction tests did not reveal there to be statistically significant differences in associations between income groups (p=0.908) and NS-SEC groups (p=0.128). Switching to active travel was associated with a $-0.71\%$ lower body fat (95% CI $-1.47\%$ to 0.05%) among children in the lowest household income group, compared with a $-0.55\%$ lower body fat (95% CI $-1.01\%$ to $-0.09\%$) among those in the highest income group. Among the economically inactive NS-SEC group, switching to an active travel mode was associated with a $-0.78\%$ lower body fat (95% CI $-1.27\%$ to $-0.29\%$) compared with a $-0.43\%$ lower body fat (95% CI $-0.78\%$ to $-0.08\%$) among the managerial/professional NS-SEC group.

Exploratory analyses of the potential role of journey duration to school in findings by SEP were performed for individuals walking or cycling to school. These found that more affluent children were more likely to travel more than 15 min each way to school. A total of 60.5% of

children in the managerial/professional NS-SEC group travelled 15 min or further compared with 51.9% of those in the not economically active NS-SEC group (adjusted OR (AOR) 1.18, 95% CI 1.04 to 1.34). A total of 60.7% of children in the highest income households travelled 15 min or more to school compared with 48.6% of those in the lowest income households (AOR 1.21, 95% CI 1.03 to 1.42. Online supplemental table 5 analyses interacting duration with mode of travel were restricted to only ages 11 and 14 years due to data availability, but interaction tests did not reveal a difference between the associations for walkers/cyclists more than versus less than 15 min away for BMI (p=0.259) and body fat (p=0.452).

## DISCUSSION

This longitudinal study followed children over a key period in the life course, and found that switching from private motorised modes of travel to school to active travel was associated with reduced adiposity. Additionally, switching to the use of public transport was associated with lower adiposity, although associations with BMI did not reach statistical significance. There was also a suggestion that these associations may be more concentrated among the most disadvantaged children, who are at greater risk of excess adiposity and relevant sequelae.

### Strengths and limitations

This study used an established and representative UK birth cohort study with high-quality data following up the same children over 8 years. We used two objective measures of adiposity and employed models that produce robust estimates of within-individual changes, which improve confidence in these results. We were able to adjust for unobserved time-invariant individual characteristics that can

bias the relationship between travel mode and adiposity. Nonetheless, there are limitations to this research which should be considered. We rely on self-reported mode of travel to school, which while it was consistent across the study period, did not capture information on multimode journeys or variation in different modes across the week. This may have introduced some misclassification in our findings. Our reliance on these self-report questions is in contrast to some other research using the MCS, which has used objective physical activity measures by accelerometers.[25] However, issues with using accelerometer data here include that is currently not available for the age 14 years sweep; sample sizes being substantially lower for this method of data collection and difficulties in ascertaining cycling to school. There was an average of 3 years between data collection points, and we do not know at what point between data collections individuals changed their travel behaviour. While we were able to consider a wide range of covariates, there may be some residual confounding from time-varying factors not considered.[26] We have included some exploratory analyses of whether there are inequalities in the time taken to get to school. For these, the distance between home and school may have served as a more accurate marker of physical activity dose, although these data were not available. There is also the potential for some confounding from other lifestyle factors as, for example, individuals who switch to more active modes may be choosing to lead a healthier lifestyle in other ways also, and changes in adiposity may be occurring via alternative mechanisms than active travel.

### Comparison with other research and practical implications

This study contributes to the evidence base by using a robust longitudinal examination of changes in travel modes and adiposity as well as specifically investigating whether there are differences across markers of SEP. We use a national sample and we were able to employ more robust methods, control for more potential confounding factors and follow children up for longer compared with many previous studies. The primary finding of this study is that switching to using an active mode of travel was linked to lower adiposity which adds to the evidence base that active forms of travel to school are linked with more favourable levels of adiposity.[11 20 27] A review in 2011 found a limited number of studies but concluded that active travel to school was associated with greater cardiorespiratory fitness and that increased physical activity is a likely pathway.[11] The second main finding of this study is that switching to public transport was linked to reduced adiposity although associations with BMI did not reach statistical significance. There has been limited consideration of the impacts of public transportation use on health, but this finding does broadly concur with a recent systematic review in adults which identified 10 relevant studies. This systematic review found that switching to public transport was associated with a −0.30 decrease in BMI, which is larger than our point estimate of −0.13, although in adults.[28] We also find no statistical evidence

of differences between SEP groups, although there was a trend towards larger effects in more deprived groups. However, we caution against overinterpretation of this finding given our restriction to only examining within-individual variation (hence reducing our statistical power) and the limited number of individuals within the study after stratifying by markers of SEP.

Nonetheless, active travel is only one potential mechanism to reduce adiposity in the population, and our findings also point to beneficial impacts of eating breakfast every day and sporting physical activity. For example, the associations identified here for switching to active forms of travel were considerably smaller than for switching to 5 or more days per week of physical activity for both BMI ($-0.22$ kg/m$^2$ vs $-0.32$ kg/m$^2$) and body fat ($-0.55\%$ vs $-1.00\%$). Similarly, switching to being physically active $\geq 5$ days per week was associated with larger changes in adiposity (eg, 1.04% of body fat). Both of these findings concur with a wealth of other evidence on the importance of physical activity and dietary behaviours, both using MCS data and international evidence.[16 29]

While we find higher levels of active travel use among more deprived groups, the wider evidence base on physical activity by markers of deprivation presents a mixed picture. Research using objective physical activity data from the MCS found that when they were aged 7 years, children in more deprived areas were more likely to be obese, but were also more likely to hit recommended levels of physical activity.[30] Recent reviews have questioned whether there are differences in physical activity by deprivation, and concluded that while more affluent children do more leisure time activity, overall differences remain small for physical activity overall and for active travel specifically.[31]

These findings highlight the potential for a shift to physically active modes of travel to play a role in reducing population-level adiposity, and potentially to reduce health inequalities. They also reinforce calls from the WHO to develop policies which promote incorporating physical activity into daily lives.[32] However, there is mixed evidence on the how best to increase levels of active travel to school, which remains a fruitful avenue for future research.[33] Additionally, determinants of active travel are known to be wide ranging including parental concerns regarding safety, attitudes of the school and wider environmental factors.[34–38] Systematic reviews have found a wide range of potential determinants, including negative associations with traffic volumes, but these results were heterogeneous due to differing contexts and measures.[39] This means that there is a need for context-specific research alongside practical actions to increase the use of physically active travel modes. These will likely need to act at a number of levels, including addressing availability of safe cycle routes, school-level policies to discourage driving and national policies to improve infrastructure. Given the well-established benefits of physical activity on outcomes other than adiposity, and the potential additional benefits of increasing active travel in the form of air

pollution and greenhouse gases, concerted action should be made to encourage the use of more active modes of travel to school.[40]

## CONCLUSION

This 7-year longitudinal study with data collected at three time points has concluded that switching to active forms of travel can have beneficial adiposity impacts at a key time in the life course. These benefits may be more apparent among disadvantaged children. This research adds to the evidence base on the health impacts of active travel and suggests that targeted structural interventions may reduce inequalities in health.

**Acknowledgements**  We would like to acknowledge the participants of the MCS.

**Contributors**  AAL and CM conceived the study. AAL, CM and TH devised the analyses. AAL conducted the analyses with the assistance of TH. AAL produced the initial draft, and all authors revised this for significant intellectual content and approve the final version.

**Funding**  This study/project is funded by the National Institute for Health Research (NIHR) School for Public Health Research (grant reference number PD-SPH-2015) as well as an NIHR Research Professorship award to CM (NIHR RP_2014-04-032).

**Disclaimer**  The views expressed are those of the author(s) and not necessarily those of the NIHR or the Department of Health and Social Care.

**Competing interests**  None declared.

**Patient consent for publication**  Not required.

**Ethics approval**  As this study involved secondary analyses of data that do not contain identifiable information, the MCS ethical approval was not required. Data collection for the MCS has ethical approval from the Yorkshire and Humber ethics committee (11/YH/0203) and further details are available from http://www.cls.ioe.ac.uk/. Participants give consent for their data to be used for research purposes.

**Provenance and peer review**  Not commissioned; externally peer reviewed.

**Data availability statement**  MCS data are available from https://ukdataservice.ac.uk/.

**ORCID iDs**
Anthony A Laverty http://orcid.org/0000-0003-1318-8439
Thomas Hone http://orcid.org/0000-0003-0703-6973
Yvonne Kelly http://orcid.org/0000-0002-2936-3994

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
