## [Reviewer comments · BMJ Open]

ARTICLE DETAILS

TITLE (PROVISIONAL)	Associations of active travel with adiposity among children and socio-economic differentials: a longitudinal study
AUTHORS	Laverty, Anthony; Hone, Thomas; Goodman, Anna; Kelly, Yvonne; Millett, Christopher

VERSION 1 – REVIEW

REVIEWER	Palma Chillón Garzón University of Granada, Spain
REVIEW RETURNED	13-Feb-2020

GENERAL COMMENTS	Dear authors, The manuscript addresses an interesting topic about the longitudinal associations between active travel and adiposity, focusing in the socio-economic differences of the population, using a large sample size of British young people. It contributes to the scientific literature regarding the benefits of active travel, and in addition, it includes more healthy lifestyle related to adiposity. There are some major and minor issues to address in order to improve the manuscript and for a better understanding. 1. Introduction a. In the 1st paragraph, it is suggested to include more international references, apart from the British references, since the problem exposed is a worldwide problematic. b. In the 2nd paragraph, the 1st sentence states that there is associations between model of travel and adiposity; however, this statement is not clear in the scientific references (in fact, the authors state this inconclusive result in the next paragraph when mentioning the previous reviews about health benefits of active travel). Consequently, it is suggesting rewriting it for avoiding contradictory information and clarifying it. c. It is suggested to include information about other healthy behaviours that are analysed in the manuscript such as physical activity, screen time or some nutrition behaviours. They must be mention is they will be analysed later. 2. Method. a. It is suggested including the statistical software used. b. Why the authors indicate that the adiposity measures are objective? It is not common to see objective to mention BMI, but it maybe correct. c. In the 2nd paragraph, it says that the report of mode of travel was referred to the longest distance. However, talking about the effect of active travel in adiposity, it maybe more relevant the time
---

	of travelling. So, if the time data is available, it is suggested using it. If not, maybe authors must remark that may be a limitation. d. In the analyses, it is suggested to specify the exact variables that were used to adjust. It says: "This, for example, could be sex...". It is suggested including the real ones. e. Regarding the adjusting variables, a relevant one is the "distance home-school" since is the 1st predictor of active travel. So, it is suggested to include it in every statistical analysis due to the high importance that it has and then may be affecting the relationship active travel vs adiposity. Although the exploratory analysis includes the distance, it should be too a main confounder in the main analysis. f. In the analysis, it is suggested including which are those "unmeasured" variables that control the analysis, and what the time-variant confounders are. g. In the exploratory analysis, it must indicate that the analysis with the time from home to school, were performed only for active travels, since it make no sense to include the distance for passive travellers. In addition, it must be clarified again in those results. 3. Results. a. Table 1 and 2. It is suggested moving the categories of the variable "NS-SEC social class" from lowest to highest, for consistency with the previous SES variable and to be same as in the appendix table 1 and 2. In addition, to reorganize the categories of these 2 variables in the appendix table 3 and 4 to be consistent. b. The appendix table 5. A title in the 1st file is lacking 4. Discussion a. The discussion would need further information about the main result, why it may happen and discuss it with other similar findings. b. Another interesting point to discuss if that, regarding the several healthy behaviours measured, which one contributes more to a better adiposity level and why c. Finally, the explanation of future studies and the practical implications should be clearly stated in the discussion.
--	---

REVIEWER	Daniel Camiletti-Moirón 1. Department of Physical Education, GALENO Research Group, School of Education Sciences, University of Cádiz, Avenida República Saharaui s/n, 11519, Puerto Real, Cádiz, Spain 2. Biomedical Research and Innovation Institute of Cádiz (INiBICA) Research Unit, Puerta del Mar University Hospital University of Cádiz, Cádiz, Spain
REVIEW RETURNED	14-May-2020

GENERAL COMMENTS	Thank you for inviting me to review this paper. The longitudinal design (including three sweeps), the valid measures of several BMI and the large sample are important strengths. Abstract 1. The authors should rewrite some acronyms of the abstract that were not named before (e.g. SEP). 2. In the results section you should report the Beta coefficients, ICs and p value properly. I would add the age range of the study sample as well. 3. Why the authors excluded the "other" response?
--

	Methods  1. How many schools were involved in the study? 2. I would add the age range of the sweeps selected. 3. Table 1 miss information regarding the values which are presented such as the percentage and the total of each variable. The table should be self-explanatory. 4. Further, to describe the sample, the authors have categorized walking and cycling as active travelers, but it would be relevant to also give percentages of walking and cycling participants separately. 5. I suggest adding information on how SEP was measured. 6. Although the authors nicely describe this in the limitations section, could the authors also include the change in % of Active /Passive commuters over the three sweeps. 7. To better understand the Longitudinal analyses models a illustration figure should be included. Results  - In the results section you should report the Beta coefficients, ICs and p value properly. Conclusion  - I would not refer to a seven-year study, I would refer to 3 sweeps in a seven-year study.
--	--

REVIEWER	Hongyan Xu Augusta University
REVIEW RETURNED	05-Jun-2020

GENERAL COMMENTS	My main concern is the analysis method. This is a longitudinal study and the authors mentions "Longitudinal regression models" without much detail. It would be more helpful to clearly spell out the details of the model, including the fixed effects and random effects.
---

REVIEWER	Konstantinos Pateras Department of Biostatistics and Research Support, Julius Center for Health Sciences and Primary Care, University Medical Center Utrecht, The Netherlands
REVIEW RETURNED	13-Jun-2020

GENERAL COMMENTS	Given my statistical background I will focus mostly on statistical and general issues. Laverty et al have conducted a retrospective longitudinal study and a cross-sectional evaluation of the last available sweep of children to seek for associations between modes of transport to school and adiposity. In my opinion the question is clear and possibly relevant given the cited literature. However, I am very concerned in regards to the small emphasis and missing details of the statistical approaches employed. Also, to ensure replication of results, proper analytical reporting of the methods is crucial, especially in the current setting were complicated analyses are being performed. Main Points
---

	1. The statistical methods are currently very non-specific and they have to be analytically reported. This can be achieved with proper statistical references to articles, specific details in the main text and the use of details on specific statistical packages and programs (I assumed that the authors used Stata...). 2. Which specific models did the authors use? Longitudinal analysis contains a large pool of proper and possibly improper models. "Longitudinal fixed-effects regression" can be very general. Even if we assume that a specific statistical model was reported, the reporting of the modeling options should also be clearer (Table 2), That could be easily done as follows. In the supplementary material the authors could provide the code for the model(s) applied. This general reporting of methods impacts directly and indirectly the results as well, see minor points below. Minor Points The authors should add units in Table 1 in description or in the first row and also elsewhere. In Table 2 the authors can add " -- " where " ref " exists to help readers. They can add reference next to the reference level as well e.g. 7 (reference) Figure 1 needs to be fixed both visually (square box, larger fonts, better resolution, one y axis or two figures 1a and 1b) and better description on the caption which now is very general. The same holds for Figure 2. What changes are the authors comparing here? Active vs. non active modes or a specific non active with active? They should be more specific here as well. The authors should be systematic. Use, + or no sign to denote positive effects and other values. Why the authors did not consider to apply a random-effects model at least in regards to specific factors such as; country? If multivariate modeling was applied given the number of variables each sub-subgroup will result with a very small number of observations. I counted between 3,000 to 36,000 children groups given different modeling. Either-way, the authors should report clearly if their modeling was multivariate, which variables they controlled for, Table 2 implies a single model on the title and multiple models on the subtitle with various adjustments. pg 7 lines 26-28. The authors claim that "Specific ethical approval was not required for this study as it involved only the secondary analyses of data that does not contain identifiable information." I guess that special ethical approval may not be required but this sentence may imply that consent during the interview phase with the caregivers was not asked for. pg. 9 lines 58-59. "Caregiver-reported beginning of growth spurts between ages 11 and 14 was associated with increased BMI (0.22 kg/m², 0.14 to 0.31)." Commas could be used here "... ,between ages 11 and 14, ...".
--	---

	pg.11 lines 47-50. Could the authors discuss this strength? I though that time-invariant characteristics could be easily accounted for in cross sectional studies.
--	--

VERSION 1 – AUTHOR RESPONSE

Reviewer: 1

Reviewer Name: Palma Chillón Garzón

The manuscript addresses an interesting topic about the longitudinal associations between active travel and adiposity, focusing in the socio-economic differences of the population, using a large sample size of British young people. It contributes to the scientific literature regarding the benefits of active travel, and in addition, it includes more healthy lifestyle related to adiposity.

There are some major and minor issues to address in order to improve the manuscript and for a better understanding.

We thank the reviewer for this positive assessment

1. Introduction

a. In the 1st paragraph, it is suggested to include more international references, apart from the British references, since the problem exposed is a worldwide problematic.

We thank the reviewer for this suggestion and have made the international nature of this problem and the evidence on it, clearer.

b. In the 2nd paragraph, the 1st sentence states that there is associations between model of travel and adiposity; however, this statement is not clear in the scientific references (in fact, the authors state this inconclusive result in the next paragraph when mentioning the previous reviews about health benefits of active travel). Consequently, it is suggesting rewriting it for avoiding contradictory information and clarifying it.

We thank the reviewer for pointing this out. We have now amended this so it is clearer that we are referring to a potential for active travel to reduce adiposity, although there is still some uncertainty about this in the literature.

c. It is suggested to include information about other healthy behaviours that are analysed in the manuscript such as physical activity, screen time or some nutrition behaviours. They must be mention is they will be analysed later.

We thank the reviewer for this suggestion and in the second paragraph of the introduction we mention that there is inequality in a range of behaviours linked to obesity, such as screen time and dietary behaviours.

2. Method.

a. It is suggested including the statistical software used.

We now mention that data was analysed with Stata 14.0

b. Why the authors indicate that the adiposity measures are objective? It is not common to see objective to mention BMI, but it maybe correct.

Body Mass Index in this study has been measured objectively by nurses. We mention this to differentiate measures from subjective (i.e. self-reported) measures of BMI

c. In the 2nd paragraph, it says that the report of mode of travel was referred to the longest distance. However, talking about the effect of active travel in adiposity, it maybe more relevant the time of travelling. So, if the time data is available, it is suggested using it. If not, maybe authors must remark that may be a limitation.

We thank the reviewer for raising this. The MCS data does not include information on distance to school and so we have noted that this as a limitation.

d. In the analyses, it is suggested to specify the exact variables that were used to adjust. It says: "This, for example, could be sex...". It is suggested including the real ones.

We thank the reviewer for pointing this out. In line with other reviewer comments we have now rewritten this section substantially. While we still give this as an example of an individual fixed effect, we now clarify that we are controlling for changes in country; highest NSSEC in household; household income; eating breakfast; frequency of eating breakfast; self-reported growth spurt; hours of TV and computer use per day; and days per week of moderate-to-vigorous physical activity.

e. Regarding the adjusting variables, a relevant one is the “distance home-school” since is the 1st predictor of active travel. So, it is suggested to include it in every statistical analysis due to the high importance that it has and then may be affecting the relationship active travel vs adiposity. Although the exploratory analysis includes the distance, it should be too a main confounder in the main analysis.

We do not have data on the distance between home and school although we do have data on the duration of journeys. As the reviewer points out below, including data on distance would mean that we would be unable to assess any potential impact of using public transport on adiposity. Additionally, this data was not collected in the age 7 questionnaire and using this would restrict our analyses to two time points. We have clearly stated that the analyses using duration are exploratory only for these reasons and do not feel that including it as a main confounder is appropriate.

f. In the analysis, it is suggested including which are those “unmeasured” variables that control the analysis, and what the time-variant confounders are.

We have now rewritten this section to clarify that we are controlling for individual level fixed effects (i.e. factors which do not change) as well as changes in country; highest NSSEC in household; household income; eating breakfast; frequency of eating breakfast; self-reported growth spurt; hours of TV and computer use per day; and days per week of moderate-to-vigorous physical activity.

g. In the exploratory analysis, it must indicate that the analysis with the time from home to school, were performed only for active travels, since it make no sense to include the distance for passive travellers. In addition, it must be clarified again in those results.

We now clarify in both the methods and the results that these exploratory analyses are performed only on individuals using physically active travel modes (walking or cycling).

3. Results.

a. Table 1 and 2. It is suggested moving the categories of the variable “NS-SEC social class” from lowest to highest, for consistency with the previous SES variable and to be same as in the appendix table 1 and 2. In addition, to reorganize the categories of these 2 variables in the appendix table 3 and 4 to be consistent.

We thank the reviewer for these suggestions and have amended these tables accordingly

b. The appendix table 5. A title in the 1st file is lacking

We thank the reviewer for pointing this out and have amended this

4. Discussion

a. The discussion would need further information about the main result, why it may happen and discuss it with other similar findings.

We have now expanded our discussion of the main finding that active modes of travel were associated with reduced adiposity in the “comparisons with other research” section. We are now clearer that the likely reason for this is physical activity. We have also restructured the discussion to be clearer about the findings and how they compare with other research.

b. Another interesting point to discuss if that, regarding the several healthy behaviours measured, which one contributes more to a better adiposity level and why

We now mention comparisons with both physical activity and eating breakfast in terms of adiposity benefits in the discussion.

c. Finally, the explanation of future studies and the practical implications should be clearly stated in the discussion.

We have now changed the penultimate paragraph of the discussion to be clearer that there is a need for context specific further research on changing travel patterns. The practical implications are that there needs to be a greater focus at a number of levels on achieving this and that a wide range of policies will be needed.

Reviewer: 2

Reviewer Name: Daniel Camiletti-Moirón

Thank you for inviting me to review this paper. The longitudinal design (including three sweeps), the valid measures of several BMI and the large sample are important strengths.

We thank the reviewer for their positive assessment

Abstract

1. The authors should rewrite some acronyms of the abstract that were not named before (e.g. SEP).

Thank you for pointing this out, we have now made this change

2. In the results section you should report the Beta coefficients, ICs and p value properly. I would add the age range of the study sample as well.

We presume that the reviewer is suggesting adding p values alongside the beta coefficients and confidence intervals. We feel that this information would be superfluous as the confidence intervals allow a greater assessment of statistical significance and uncertainty. We now provide more detail in the methods section of the paper on the age range of participants.

3. Why the authors excluded the “other” response?

Space constraints preclude us from addressing this point in the abstract but we are now clearer in the methods that we exclude the “other” category as this is likely to be a heterogeneous mix of travel modes.

Methods

1. How many schools were involved in the study?

Sampling for the MCS is not based on schools attended by the children and this information is not available in the data we have. The sample of children however comes from several thousand schools, is representative of those in the four nations of the UK

2. I would add the age range of the sweeps selected.

We have now added in the methods that in the fourth sweep of MCS children were aged between 6 and 8 years old. At the fifth sweep this was between ages 10 and 12 years and at the sixth sweep was between 13 and 15 years.

3. Table 1 miss information regarding the values which are presented such as the percentage and the total of each variable. The table should be self-explanatory.

We have now added these details to table 1

4. Further, to describe the sample, the authors have categorized walking and cycling as active travelers, but it would be relevant to also give percentages of walking and cycling participants separately.

We now clarify that there were low numbers of cyclists which necessitated combining walking and cycling into an overall active group. There were 76 participants cycling to school at age 7, 187 at age 11 and 147 at age 14.

5. I suggest adding information on how SEP was measured.

We use two markers of SEP and these are both based on self-report using validated questions. These were household income group in five groups, (equivalised based on the Organisation for Economic Co-operation and Development) and household occupational social class measured using the National Statistics Socio-Economic Classification (NS-SEC; categorised as Not economically active; Lower; Intermediate; Managerial/Professional).

6. Although the authors nicely describe this in the limitations section, could the authors also include the change in % of Active /Passive commuters over the three sweeps.

We thank the reviewer for this comment. We have clarified this by adding detail to the abstract about the overall levels of change. We also mention this in the methods as well as the fact that switching travel modes was more common between the ages of 11 and 14, than between 7 and 11 years.

7. To better understand the Longitudinal analyses models a illustration figure should be included.

In line with comments from other reviewers we have now substantially changed our description of the analyses performed and these should now be considerably clearer.

Results

- In the results section you should report the Beta coefficients, ICs and p value properly.
In the interests of space we feel that it would be repetitive and unnecessary to report p values alongside the beta coefficients and confidence intervals which we report.

Conclusion

- I would not refer to a seven-year study, I would refer to 3 sweeps in a seven-year study.
We have made this change

Reviewer: 3

Reviewer Name: Hongyan Xu

My main concern is the analysis method. This is a longitudinal study and the authors mentions "Longitudinal regression models" without much detail. It would be more helpful to clearly spell out the details of the model, including the fixed effects and random effects.

We have now substantially rewritten the methods section to give more detail and clarity on the analytical approach. We are now clearer (for example in the abstract) that we have used longitudinal (panel) regression models and that these have adjusted for individual level fixed effects. We now clarify that we implemented the model using Stata and the xtreg commands. We are using a model with individual level fixed effects and we additionally control for changes in country; highest NSSEC in household; household income; eating breakfast; frequency of eating breakfast; self-reported growth spurt; hours of TV and computer use per day; and days per week of moderate-to-vigorous physical activity. We do not employ models which include random effects as we have designed the study to examine individual level changes in travel mode and adiposity. We did however, perform the hausman test of random vs. fixed effect models, which indicated ($p < 0.001$) that fixed effects were superior.

Reviewer: 4

Reviewer Name: Konstantinos Pateras

Given my statistical background I will focus mostly on statistical and general issues.

Laverty et al have conducted a retrospective longitudinal study and a cross-sectional evaluation of the last available sweep of children to seek for associations between modes of transport to school and adiposity. In my opinion the question is clear and possibly relevant given the cited literature. However, I am very concerned in regards to the small emphasis and missing details of the statistical approaches employed. Also, to ensure replication of results, proper analytical reporting of the methods is crucial, especially in the current setting where complicated analyses are being performed.

We address these specific points below

Main Points

1. The statistical methods are currently very non-specific and they have to be analytically reported. This can be achieved with proper statistical references to articles, specific details in the main text and the use of details on specific statistical packages and programs (I assumed that the authors used Stata...).

In line with other comments we have now rewritten this section to be much clearer about our models and the fact that we used xtreg within stata. The main methodological sources for our choices are Wooldridge JM. "Econometric analysis of cross section and panel data" and Rabe-Hesketh S, Skrondal A. "Multilevel and longitudinal modeling using Stata." We now reference both of these

2. Which specific models did the authors use? Longitudinal analysis contains a large pool of proper and possibly improper models. "Longitudinal fixed-effects regression" can be very general. Even if we assume that a specific statistical model was reported, the reporting of the modeling options should also be clearer (Table 2), That could be easily done as follows. In the supplementary material the authors could provide the code for the model(s) applied. This general reporting of methods impacts directly and indirectly the results as well, see minor points below.

In line with the comments below we have improved the presentation of table 2. We are now clearer that we are using a model with individual level fixed effects and have reported the specific variables we controlled for.

Minor Points

The authors should add units in Table 1 in description or in the first row and also elsewhere.
Thank you, we have added these for clarity

In Table 2 the authors can add " -- " where " ref " exists to help readers. They can add reference next to the reference level as well e.g. 7 (reference)

We believe that "ref" is sufficient here and more understandable. We have added the detail that this stands for reference category

Figure 1 needs to be fixed both visually (square box, larger fonts, better resolution, one y axis or two figures 1a and 1b) and better description on the caption which now is very general. The same holds for Figure 2. What changes are the authors comparing here? Active vs. non active modes or a specific non active with active? They should be more specific here as well.

We apologise for this error which was caused by omitting more detailed captions upon submission. We are now clearer in the captions that these represent results from our regression models and they give average effect of switching to using active transport from any other mode. We have also improved presentation of these figures

The authors should be systematic. Use, + or no sign to denote positive effects and other values.

We are now consistent in using no sign for positive values in the tables although we use "+" for clarity in the text

Why the authors did not consider to apply a random-effects model at least in regards to specific factors such as; country? If multivariate modeling was applied given the number of variables each sub-subgroup will result with a very small number of observations. I counted between 3,000 to 36,000 children groups given different modeling. Either-way, the authors should report clearly if their modeling was multivariate, which variables they controlled for, Table 2 implies a single model on the title and multiple models on the subtitle with various adjustments.

We have chosen a fixed effect model to concentrate on individual level change in travel modes and adiposity, for which random effects would not be appropriate. We additionally performed the Hausman test and found $p < 0.001$, indicating that the fixed effect model was superior. We are now clear in the rewritten methods section that models and multivariate and we are clear what these control for. We have made it clearer in the footnotes to table 2 that there is one model for each of the two outcomes.

pg 7 lines 26-28. The authors claim that "Specific ethical approval was not required for this study as it involved only the secondary analyses of data that does not contain identifiable information." I guess that special ethical approval may not be required but this sentence may imply that consent during the interview phase with the caregivers was not asked for.

We now clarify that participants gave consent for their data to be used for research purposes

pg. 9 lines 58-59. "Caregiver-reported beginning of growth spurts between ages 11 and 14 was associated with increased BMI (0.22 kg/m², 0.14 to 0.31)." Commas could be used here "... ,between ages 11 and 14, ...".

We are unclear what the reviewer would prefer here although have been through the paper thoroughly for grammar and punctuation and made various improvements.

pg.11 lines 47-50. Could the authors discuss this strength? I thought that time-invariant characteristics could be easily accounted for in cross sectional studies.

We thank the reviewer for pointing this out and we have now removed the mention of cross-sectional studies here

VERSION 2 – REVIEW

REVIEWER	Palma Chillón University of Granada, Spain
REVIEW RETURNED	15-Oct-2020

GENERAL COMMENTS	The authors have addressed the previous comments and the manuscript has been improved. In addition, there are two previous comments that I would like to reemphasize since they could improved and clarify the method. 1. Regarding the adiposity measures, in method, the authors answer that Body Mass Index was measured objectively because it was performed by nurses. It should be better explained, because it may mean that only when nurses perform it is objective; if a teacher performs the measurement should be subjective? It is suggested providing a deeper explanation about why it is objective or rethink if it may be not objective regarding the real meaning of objectively (i.e., Dexa). 2. In method, the variable "duration of the journey" was measured and used for exploratory analysis. It was not used in the main analysis, and the authors state that "we do not feel that including it as a main confounder is appropriate". I highlight the importance of having this variable (which may be a close measure to distance home-school) to be used and I emphasize on including it in the main analysis in spite that, as authors indicate, it was not collected in the age 7 questionnaire. It is suggested including it in the analysis that is possible, since new results and insights can overcome.
--

REVIEWER	Daniel camiletti-Moirón University of Cadiz (Spain)
REVIEW RETURNED	04-Nov-2020

GENERAL COMMENTS	The authors have responded clearly and properly to the comments done for the reviewer.
--

REVIEWER	Hongyan Xu Augusta University
REVIEW RETURNED	22-Oct-2020

GENERAL COMMENTS	The authors have improved on the description of the statistical methods over the previous manuscript. In the response, the author claimed to have performed the hausman test of random vs. fixed effect models. The result should be included in the manuscript to justify the use of fixed models.
---

REVIEWER	Konstantinos Pateras University Medical Center Utrecht, The Netherlands
REVIEW RETURNED	12-Oct-2020

GENERAL COMMENTS	The authors did a well clarifying most of my comments. Still, in many places the grammar and syntax could be considerably improved to aid readers. Please consider letting a native speaker to proofread your text. In at least one occasion (point 3) the syntax could lead to misinterpretations. For instance but also elsewhere, 1. In methods, "This we controlled for changes in country; highest ..." Change to either active or passive e.g. Confounders we controlled include, ... Or Controlled confounders include 2. Double "the" near the end of the above paragraph. 3. One paragraph below, "First, these models were used in two separate models to assess the associations between active transport and BMI and percentage body fat. Second to assess differences across our two markers of SEP we used interaction tests to assess if differences were statistically significantly different and present stratified analyses." The phrasing these models were used in two separate models is confusing. Do the author mean that they applied two versions of these model? Then the remaining paragraph needs to be checked for syntax and grammar. Significantly does not take "ly' at the end. Present is used as a noun or as a verb in the end? If it is a verb it should read "... and we presented stratified analyses". Use commas to make easier for the reader to follow. In the current form, the sentence can be read in many different ways. 4. " Analyses were conducted using Stata 14.0 and we implemented using the xtreg package for longitudinal analysis." And we implemented them using the xtreg package.. Or "...using Stata via the xtreg package...". 5. Page 11, "although this data was not available." Data were not available
--

VERSION 2 – AUTHOR RESPONSE

Reviewer: 1

Reviewer Name: Palma Chillón

Institution and Country: University of Granada, Spain Please state any competing interests or state 'None declared': None declared

The authors have addressed the previous comments and the manuscript has been improved. In addition, there are two previous comments that I would like to reemphasize since they could improved and clarify the method.

We are pleased that the reviewer agrees that the paper is improved.

1. Regarding the adiposity measures, in method, the authors answer that Body Mass Index was measured objectively because it was performed by nurses. It should be better explained, because it may mean that only when nurses perform it is objective; if a teacher performs the measurement should be subjective? It is suggested providing a deeper explanation about why it is objective or rethink if it may be not objective regarding the real meaning of objectively (i.e., Dexa).

We have clarified that we refer to these measures as objective in that they are not based on self-report. We have added the sentence "This provides objective data on BMI, rather than relying on participants self-reporting their height and weight."

2. In method, the variable "duration of the journey" was measured and used for exploratory analysis. It was not used in the main analysis, and the authors state that "we do not feel that including it as a main confounder is appropriate". I highlight the importance of having this variable (which may be a close measure to distance home-school) to be used and I emphasize on including it in the main analysis in spite that, as authors indicate, it was not collected in the age 7 questionnaire. It is suggested including it in the analysis that is possible, since new results and insights can overcome. **We respectively disagree and not that the editor feels this point has been adequately addressed.**

Reviewer: 2

Reviewer Name: Daniel camiletti-Moirón

Institution and Country: University of Cadiz (Spain) Please state any competing interests or state 'None declared': None declared

The authors have responded clearly and properly to the comments done for the reviewer.

Thank you

Reviewer: 3

Reviewer Name: Hongyan Xu

Institution and Country: Augusta University, USA Please state any competing interests or state 'None declared': None declared.

The authors have improved on the description of the statistical methods over the previous manuscript. In the response, the author claimed to have performed the hausman test of random vs. fixed effect models. The result should be included in the manuscript to justify the use of fixed models.

We are pleased that the reviewer agrees that the paper is improved. We have clarified the use of the Hausman test with the addition of this sentence "We performed the hausman test of random vs. fixed effect models, which indicated ($p < 0.001$) that fixed effects were superior."

Reviewer: 4

Reviewer Name: Konstantinos Pateras

Institution and Country: University Medical Center Utrecht, The Netherlands Please state any competing interests or state 'None declared': None declared

The authors did a well clarifying most of my comments.

We are pleased that the reviewer agrees the paper is improved.

Still, in many places the grammar and syntax could be considerably improved to aid readers. Please consider letting a native speaker to proofread your text. In at least one occasion (point 3) the syntax could lead to misinterpretations. For instance but also elsewhere,

We have now been through the text in detail for grammar and syntax.

1. In methods, "This we controlled for changes in country; highest ..."

Change to either active or passive e.g. Confounders we controlled include, ... Or Controlled confounders include

We have made this change

2. Double "the" near the end of the above paragraph.

We have made this change

3. One paragraph below, "First, these models were used in two separate models to assess the associations between active transport and BMI and percentage body fat. Second to assess differences across our two markers of SEP we used interaction tests to assess if differences were statistically significantly different and present stratified analyses."

The phrasing these models were used in two separate models is confusing. Do the author mean that they applied two versions of these model? Then the remaining paragraph needs to be checked for syntax and grammar. Significantly does not take 'ly' at the end. Present is used as a noun or as a verb in the end? If it is a verb it should read "... and we presented stratified analyses". Use commas to make easier for the reader to follow. In the current form, the sentence can be read in many different ways.

We have changed this and it now reads "First, this model was used for the two separate outcomes of BMI and percentage body fat. Second to assess differences across our two markers of SEP we used interaction tests to assess if differences were statistically different and performed stratified analyses. All analyses used cluster robust standard errors to account for heteroskeascity and autocorrelation, and employed survey weights to correct for differential response rates between groups and over time [22][23]. We performed the hausman test of random vs. fixed effect models, which indicated (p<0.001) that fixed effects were superior . Analyses were conducted using Stata 14.0 and implemented using the xtreg package for longitudinal analysis."

4. " Analyses were conducted using Stata 14.0 and we implemented using the xtreg package for longitudinal analysis."

And we implemented them using the xtreg package..

Or "...using Stata via the xtreg package..."

We have clarified this sentence by removing the middle "we" so it now reads "Analyses were conducted using Stata 14.0 and implemented using the xtreg package for longitudinal analysis."

5. Page 11, "although this data was not available."

Data were not available

We have made this change

VERSION 3 – REVIEW

REVIEWER	Hongyan Xu Augusta University, United States
REVIEW RETURNED	03-Dec-2020
GENERAL COMMENTS	This is a revision and the authors have addressed my concerns sufficiently.